# Capsaicin and Zinc Signalling Pathways as Promising Targets for Managing Insulin Resistance and Type 2 Diabetes

**DOI:** 10.3390/molecules28062861

**Published:** 2023-03-22

**Authors:** Parisa Vahidi Ferdowsi, Kiran D. K. Ahuja, Jeffrey M. Beckett, Stephen Myers

**Affiliations:** 1School of Health Sciences, College of Health and Medicine, University of Tasmania, Newnham Drive, Launceston, TAS 7248, Australia; parisa.vahidiferdowsi@utas.edu.au (P.V.F.); kiran.ahuja@utas.edu.au (K.D.K.A.); jeffrey.beckett@utas.edu.au (J.M.B.); 2Children’s Cancer Institute, Lowy Cancer Research Centre, C25/9 High St, Kensington, NSW 2750, Australia

**Keywords:** type 2 diabetes, insulin resistance, glucose metabolism, capsaicin, zinc, calcium signalling, CAMKK, CREB, TORC

## Abstract

The global burden of type 2 diabetes (T2DM) has led to significant interest in finding novel and effective therapeutic targets for this chronic disorder. Bioactive food components have effectively improved abnormal glucose metabolism associated with this disease. Capsaicin and zinc are food components that have shown the potential to improve glucose metabolism by activating signalling events in the target cells. Capsaicin and zinc stimulate glucose uptake through the activation of distinct pathways (AMPK and AKT, respectively); however, calcium signal transduction seems to be the common pathway between the two. The investigation of molecular pathways that are activated by capsaicin and zinc has the potential to lead to the discovery of new therapeutic targets for T2DM. Therefore, this literature review aims to provide a summary of the main signalling pathways triggered by capsaicin and zinc in glucose metabolism.

## 1. Introduction

Diabetes is the most common metabolic disorder and is a major global health issue [1,2]. In 2015, the number of people with diabetes was 415 million worldwide. The International Diabetes Federation estimates that this number will reach 642 million by 2040 [1]. The predominant forms of diabetes are type 1 diabetes mellitus (T1DM), type 2 diabetes mellitus (T2DM), and gestational diabetes [3]. T2DM, the most common type of diabetes, mainly occurs in adults aged 20–79 years old and is associated with insulin resistance (IR) [4,5].

T2DM is the result of genetic and/or environmental factors [6]. The worldwide increase in high-calorie food consumption and the prevalence of a sedentary lifestyle are two important environmental factors behind the universal rise in T2DM in recent years [2]. Long-term and uncontrolled hyperglycaemia leads to the development of IR, T2DM, and associated complications [6]. Diabetes complications include macrovascular disorders, such as coronary artery disease, peripheral arterial disease, and stroke, as well as microvascular diseases, including diabetic neuropathy, diabetic nephropathy, and diabetic retinopathy, and are the major cause of morbidity and mortality worldwide [7]. In Australia, in the year 2000, about 1 million people had diabetes, and this number is predicted to reach 2–2.9 million by 2025. Diabetes is a costly disease at both the individual and social levels [7,8,9,10,11].

IR is identified as an impaired response of the target tissues, primarily adipose tissue and skeletal muscle to insulin-stimulated glucose uptake [12]. The hallmarks of impaired insulin sensitivity in target tissues include a reduction in insulin-stimulated glucose uptake in skeletal muscle, impaired insulin-mediated inhibition of hepatic glucose production in the liver, and a decrease in lipolysis inhibition in adipose tissue, all contributing to high blood glucose levels [13]. Continued and uncontrolled hyperglycaemia has serious negative long-term effects, including cardiovascular disorders, neuropathy, stroke, and nephropathy [14]. IR manifests up to 10 to 15 years before the development of T2DM, when pancreatic β-cells are no longer able to compensate for the decreased insulin sensitivity and impaired glucose tolerance in target tissues [15]. This period provides a “window of opportunity” for lessening T2DM progression using novel and effective pharmacological approaches.

Maintaining normal blood glucose concentrations requires an accurate balance of glucose utilisation and endogenous glucose production by liver or dietary glucose intake [16]. Peripheral glucose metabolism in the liver, adipose tissue, and skeletal muscle plays an essential role in maintaining glucose homeostasis. While the liver is the major tissue controlling blood glucose levels, skeletal muscle glucose is an important peripheral site for glucose uptake in the body [17]. Preserving blood glucose metabolism is vital for human health, and its dysregulation is associated with serious health issues [18].

Current antidiabetic agents induce various side effects, such as nausea, bloating, diarrhoea, and risk of hypoglycaemia, which suggest limitations in existing pharmacotherapy and a need for the development of more effective therapies [19]. Many food ingredients have been shown to benefit human health by affecting a diverse range of tissues in the body. Recently, the bioactive food components that promote glucose metabolism by mediating signalling pathways involved in glucose uptake have received much consideration, as they may offer potential alternative therapies for diabetes [20,21,22]. Emerging research on cell signalling pathways in glucose metabolism activated by specific nutrients and food components has highlighted two dietary molecules, capsaicin and zinc, as potential therapeutic agents in glucose metabolism improvement in these disorders.

Capsaicin (8-methyl-N-vanillyl-6-none amid) is a phenolic active compound of chilli peppers responsible for their beneficial impacts on the regulation of glucose homeostasis [23,24,25,26]. Capsaicin induces its beneficial effects on glucose metabolism through different manners, including by elevating insulin sensitivity, reducing inflammatory factors, increasing fatty acid oxidation, stimulating insulin secretion from pancreatic β cells, and activating signalling molecules involved in glucose homeostasis [23,27,28].

Similarly, studies have shown the antidiabetic effects and advantageous impacts of zinc on glucose metabolism [29,30,31,32,33]. Normal zinc homeostasis is essential to maintain normal cellular function, and zinc deficiency is associated with many complications, including obesity, glucose intolerance, IR, and T2DM [34,35]. Zinc research demonstrates the diverse role of this ion in the control of glucose metabolism. This includes stimulation of insulin biosynthesis and secretion from pancreatic β cells, improvement in insulin sensitivity, and activation of glucose metabolism signalling events [15,36,37].

Due to the critical role of signalling pathways in the control of glucose metabolism, several studies have been conducted to understand signalling events leading to the regulation of glucose homeostasis [20,38,39]. Most of these studies investigated insulin-induced signalling pathways and their impact on normal glucose homeostasis, as well as their dysregulation in IR and T2DM [36,40]. However, numerous signalling pathways control glucose metabolism, independent of insulin [25,28,33]. Research on the insulin-independent pathways is promising, as such studies may provide alternative mechanisms to improve glucose metabolism when there is a defect in insulin synthesis or action associated with IR and T2DM [41].

While this research suggests several advantageous effects of capsaicin and zinc on the control of glucose homeostasis, the molecular mechanisms and subsequent pathways are not completely understood. Capsaicin has been demonstrated to regulate glucose metabolism through calcium/calmodulin-dependent protein kinase 2 (CAMKK2) (Throughout this review, the terms used in human systems are fully capitalized. For terms related to other systems, only the first letter is capitalized)/AMP-activated protein kinase (AMPK) signalling without having any impact on the activation of insulin signalling proteins [28,42]. In contrast, zinc improves glucose homeostasis by the activation of insulin signalling kinases such as protein kinase B (AKT), Src homology region 2 (SHP2; also known as SH2 domain-containing protein tyrosine phosphatase 2), and extracellular signal-regulated kinase 1/2 (ERK1/2), independent of insulin [29]. Although the pathways activated by capsaicin and zinc seem to be distinct, studies show that both stimulate calcium flux, which could potentially be a common pathway between capsaicin and zinc in glucose metabolism [23,28,43,44].

Calcium is a vital second messenger responsible for a wide range of cellular processes, including glucose homeostasis [45]. Additionally, dysregulated calcium signalling in IR and T2DM confirms the crucial role of this cation in normal glucose homeostasis [45]. Calcium signal transduction stimulates activation of the insulin signalling pathway, phosphoinositide 3-kinases (PI3K)/AKT, and CAMKK/AMPK signalling, which make this cation an important element in glucose metabolism [46,47]. CAMKK2 is activated downstream of calcium flux in metabolically active tissues such as skeletal muscle and is known as a potent regulator of whole-body glucose metabolism [48]. Moreover, calcium signalling activates transcription factors involved in glucose metabolism [49]. Among these transcription factors, cAMP response element binding protein (CREB) and its coactivators, a transducer of regulated CREB-binding proteins (TORCs), have emerged as the key transcriptional element in the regulation of glucose metabolism [49,50]. Activation of these proteins eventually stimulates glucose uptake by controlling the expression of genes responsible for stimulation of glucose transporter type 4 (GLUT4) translocation to the cell membrane and glucose uptake [49].

This review discusses the significance of glucose metabolism and disorders associated with abnormal glucose homeostasis and highlights signalling pathways activated by capsaicin and zinc in the regulation of glucose homeostasis with a focus on calcium signal transduction. In the end, we debate calcium signalling as a possible common pathway between capsaicin and zinc, which could potentially be targeted as a novel therapeutic strategy for IR and T2DM.

## 2. Glucose Metabolism

Owing to its high molecular weight, glucose cannot freely diffuse through the cell membrane. Thus, insulin regulates glucose transfer across the cell membrane through glucose transporter proteins (GLUTs). Any impairment in the function or expression of GLUTs can lead to insulin resistance [51]. For example, it has been demonstrated that GLUT1 and GLUT2 are involved in glucose homeostasis and may contribute to IR and T2DM. In particular, GLUT4 is the major glucose transporter protein in adipose tissue and skeletal muscle. There are several upstream signalling events in GLUT4 translocation stimulated by insulin binding to its receptor on the membrane of target cells [36]. The extracellular α subunit of the insulin receptor undergoes a conformational change upon binding with insulin, leading to the facilitation of ATP binding to the β subunit of the receptor [52]. This triggers the phosphorylation of intracellular substrate proteins and insulin-responsive substrates (IRSs) and further mediates insulin-mediated downstream signalling events, leading to GLUT4 movement and glucose uptake (as depicted in Figure 1) [53]. The phosphorylation of IRS proteins triggers the activation of phosphatidylinositol 3 kinases (PI3K). PI3K phosphorylates phosphatidylinositol-4,5-bisphosphate (PIP2), leading to the formation of phosphatidylinositol-3,4,5-trisphosphate (PIP3). AKT binding to PIP3 at the cell membrane phosphorylates and activates AKT [54]. AKT activation is involved in several cellular functions, the most important of which is its role in insulin-mediated GLUT4 translocation to the cell membrane and glucose uptake [55]. The detailed downstream signalling events whereby AKT stimulates glucose uptake are still unknown; however, the AKT substrate of 160 kDa (AS160) has been established as an important stimulator of GLUT4 retention. Phosphorylation of AS160 blocks its GTPase-activating protein (GAP) activity; subsequently, the GTP form of a Rab(s) needed for GLUT4 translocation is elevated, triggering GLUT4 movement to the cell membrane [54]. Figure 1 demonstrates insulin signal transduction-induced glucose uptake in skeletal muscle.

Although the PI3K/AKT pathway activated by insulin plays a crucial role in cellular metabolism and its dysregulation is related to many diseases, namely IR and T2DM [56], other extracellular stimuli such as muscle contraction and bioactive food components also mediate glucose uptake in skeletal muscle in an insulin-independent manner [28,29,57,58].

### Type 2 Diabetes Mellitus and Insulin Resistance

T2DM is a major metabolic disorder that is characterised by IR and a persistent increase in blood glucose levels [59]; its development involves pathological changes over a long period (Figure 2) with numerous serious complications.

IR manifests in peripheral tissues through various mechanisms [36,60,61]. IR causes mitochondrial dysfunction, endoplasmic reticulum (ER) stress, inflammation, and high levels of reactive oxygen species (ROS), which negatively affect insulin secretion from pancreatic β cells and insulin sensitivity [62,63,64,65]. An increase in non-esterified fatty acid (NEFA) levels in plasma also leads to continuous loss of pancreatic β-cell function and the development of IR [64]. Elevation in plasma NEFA levels activates a serine/threonine kinase cascade, leading to serine/threonine phosphorylation of IRS-1 and IRS-2, which reduces activation of PI3K, subsequently eliminating its downstream signalling events in target tissues, namely skeletal muscle [66]. Conversely, a decrease in plasma NEFA levels has been shown to improve peripheral insulin uptake and glucose metabolism [66].

Skeletal muscle plays an essential role in glucose uptake and is one of the primary sites of IR [40,57]. In insulin-resistant skeletal muscle, insulin-mediated glucose disposal is markedly impaired as a result of chronic inflammation [17]. For example, defects in fatty acid metabolism lead to excessive free fatty acid (FFA), which, in turn, elevates inflammation associated with skeletal muscle insulin resistance [67,68]. Fatty acid metabolism disorder-induced elevation in FFA-derived active metabolites such as diacylglycerol and ceramide activates protein kinase C isoforms, c-Jun N-terminal kinases, IκB kinase β, and nuclear factor, which, in turn, stimulate inflammatory pathways [69]. Animal models and human epidemiological studies emphasise the important role of inflammation in the development of IR through the activation of several inflammatory responses [70]. As such, differentiated cultured myocytes isolated from insulin-resistant subjects were shown to secret more inflammatory cytokines such as TNF-α compared with the control group [67]. Moreover, TNF-α knockout in mouse models of insulin resistance leads to a tolerance to the development of IR in these animals. Inflammatory mediators including TNF-α also diminish insulin sensitivity in skeletal muscle by disrupting insulin signalling pathways through a considerable reduction in IRS-1 tyrosine phosphorylation, PI3K activity, AKT phosphorylation, and GLUT4 translocation [67,68].

Extensive research on IR and T2DM has improved our knowledge of the pathogenesis of these disorders, which led to the development of therapeutic agents [71]. However, the side effects associated with current antidiabetic medicines indicate a need for the development of more efficient and safer treatments for IR and T2DM.

## 3. Chilli Peppers and Glucose Metabolism

Previous studies have shown the beneficial effects of chilli peppers on the improvement of glucose metabolism [23,24]. Regular consumption of chilli-containing foods has been reported to reduce hyperglycaemia and hyperinsulinemia with no impact on fasting plasma glucose and insulin levels in healthy individuals [24]. Similarly, acute chilli consumption improves postprandial plasma glucose levels and increases the metabolic rate within 30 min [72]. However, some other groups found no effect of chilli supplementation [73]. Studies in animal models have also reported on the positive impacts of chilli pepper administration on glucose homeostasis [18,74,75]. Capsaicin is known to mediate its effects through the activation of the transient receptor potential cation channel subfamily V member 1 (TRPV1) [23].

### Capsaicin-Induced Signalling Pathways in Glucose Metabolism

TRPV1 is a member of the transient receptor potential (TRP) family of cation channels with six putative transmembrane domains and a calcium-permeable region [76,77,78]. TRPV1 is widely expressed in metabolically active tissues, which makes this channel a potentially important therapeutic target in metabolic disorders such as IR and T2DM [23,79,80,81,82,83]. Capsaicin can permeate the plasma membrane and bind to the vanilloid-binding pocket of the TRPV1. This causes a conformational change in the channel and the opening of a protein pore, permitting calcium from the extracellular space to enter the cytoplasm, which can regulate many signalling pathways in the cells [23]. Capsaicin can also pass into the cytosol and activate TRPV1 channels on the endoplasmic reticulum (ER) [75].

Several in vivo and in vitro studies have been conducted to discover the mechanism of action of capsaicin in the regulation of glucose metabolism; however, the number and nature of capsaicin-stimulated pathways are still unknown; therefore, more research is needed. Animal studies have demonstrated some mechanisms by which TRPV1 activation by capsaicin improves glucose homeostasis. These include the reduction in inflammatory factors, stimulation of insulin secretion, and activation of signalling molecules involved in glucose uptake [23,25,27,28,84,85,86]. For example, dietary capsaicin has been shown to improve glucose metabolism through the activation of the TRPV1 channel by lowering inflammatory cytokines and NF-κB and increasing GLUT4 expression in insulin-resistant mice [23]. Similarly, the consumption of a capsaicin-containing diet reduces obesity, fasting glucose levels, and inflammatory markers in the adipose tissue and liver of high-fat-diet mice [87]. Moreover, capsaicin induces insulin secretion in wild-type mice, which was shown to be inhibited in Trpv1 knockout mice [25].

In vitro research has also shed light on the mechanism of action of capsaicin in glucose metabolism [23,28,42]. Capsaicin increases glucose metabolism in an insulin-independent manner in skeletal muscle cells [23,28]. As previously mentioned, activation of the plasma membrane and ER TRPV1 channels by capsaicin causes a calcium flux to the cytosol from either the extracellular space or intracellular stores [88]. Elevated cytosolic calcium mediates its signal transduction through an intracellular multifunctional protein, calmodulin (CAM). Calcium binding to CAM induces a structural modification in CAM and forms a calcium/calmodulin complex [89]. This complex controls the activity of several enzymes and proteins, including but not limited to transcription factors, phosphatases, and kinases [88]. Calcium/calmodulin-dependent protein kinases (CAMKs) and AMPK, the key regulators of glucose metabolism, are the main multifunctional protein kinases activated downstream of the calcium/calmodulin complex and in response to calcium levels increasing in the cells [48]. Figure 3 demonstrates the CAMK signalling pathway and the downstream events following TRPV1 activation by capsaicin in skeletal muscle. Among CAMKs, CAMK2 activation has been shown to improve glucose metabolism. It has been demonstrated that this kinase mediates glucose homeostasis by increasing insulin secretion from pancreatic β cells, controlling peripheral tissue insulin sensitivity, and stimulating the signalling events leading to glucose uptake [48,88]. Activation of CAMK2 and its downstream kinase, AMPK, triggers the activation of transcription factors including the activating transcription factor-1 (ATF-1), CAAT enhancer-binding protein-β (c/EBPβ), serum response factor (SRF), and CREB, which, in turn, mediate the expression of glucose metabolism genes [48,90].

While increasing evidence supports the beneficial role of calcium signal transduction induced by capsaicin in glucose uptake, the downstream signalling events by which calcium mediates GLUT4 translocation to the cell membrane and glucose uptake have not been fully studied. Given the apparent effects of capsaicin on glucose homeostasis and because of the crucial role of calcium signal transduction in glucose metabolism, studying the calcium signalling mediated by capsaicin could be an attractive therapeutic target to combat hyperglycaemia linked with IR and T2DM.

## 4. Relationship between Zinc and Zinc Transporters and Glucose Metabolism

Zinc is an essential micronutrient that is found in many food groups, including meat, fish, legumes, and dairy. In humans, zinc is the second most abundant trace metal in the body and can be found in all tissues [91]. Zinc is crucial for human survival, owing to its critical role in metabolic homeostasis and whole-body physiology [34,92]. The human body contains approximately 2 g of zinc, of which only 0.1% is present in plasma, with the remainder located within the cells [93]. Zinc delivery to cells, as well as its intracellular distribution, is tightly regulated by a family of proteins that is responsible for this ion’s influx and efflux [94]. The expression and cellular distribution of zinc transporters are predominately regulated by changes in extracellular and intracellular zinc concentrations [95]. In mammals, there are 24 different zinc transporter genes responsible for zinc influx or efflux. Table 1 highlights eight of these transporters that are involved in glucose metabolism. There are two main families of zinc transporters: the zinc importers (SLC39/ZIPs) and the zinc exporters (SLC30/ZnTs) [71].

Zinc plays a critical role in the formation, crystallization, and stimulation of insulin secretion from pancreatic β cells [100]. Zinc also has also been shown to inhibit glucagon secretion in response to high glucose concentrations, which improves glucose metabolism [92]. Many studies report the association of low plasma zinc levels in association with the development of diabetes [34,92,101]. Zinc plasma levels are significantly lower in diabetic patients than healthy individuals [102]. Zinc deficiency caused by reduced dietary zinc intake, inadequate zinc absorption, and increased zinc losses is associated with an increase in oxidative and ER stress, pancreatic β-cell dysfunction, insulin secretion impairment, glucose intolerance, and the development of IR and T2DM [31,92,103]. However, zinc supplementations effectively improve glucose metabolism [100,104,105]. Animal studies show that zinc supplementation lowers fasting blood glucose and insulin levels in mice [106]. Meta-analysis studies also demonstrate that zinc supplementation lowers fasting and 2 h postprandial blood glucose levels by improving inflammation status through the downregulation of inflammatory markers including hs-CRP, cytokines, and oxidative stress in diabetic patients [107,108]. Moreover, zinc supplementation elevates insulin content in pancreatic β cells, which attenuates fasting hyperglycaemia and hyperinsulinemia in diabetic mice after 4 weeks [32]. Conversely, a study demonstrated that consumption of zinc supplements did not affect fasting glucagon concentration in controlled diabetic patients with normal zinc status [109].

### Zinc-Induced Signalling Pathways in Glucose Metabolism

Maintenance of cytosolic zinc levels regulated by zinc transporters mediates cellular signalling that is amendable to glucose homeostasis and therefore has implications for IR and T2DM [29,94]. Several in vivo and in vitro studies have shown the insulin-like effects of zinc in glucose homeostasis [15,29,104,105,110]. A meta-analysis of clinical trials showed that zinc supplementation causes a significant reduction in plasma glucose concentrations in diabetic patients [111]. Similarly, cell-based studies confirm the insulin-mimetic impact of zinc on glucose metabolism [15,29,104]. Zinc was shown to enhance glucose uptake by regulation of AKT, GSK3β, and S6K1 in adipocytes [104]. Moreover, it has been demonstrated that zinc induces activation of insulin receptor β, AKT, ERK1/2, and SHP independent of insulin in myotubes, which leads to glucose uptake in these cells [29,110]. Furthermore, insulin receptor inhibition (HNMPA-(AM)3) in C2C12 skeletal muscle cells diminishes zinc-induced activation of AKT, which suggests that this cation potentially acts through a functional insulin receptor [29]. Along with the stimulation of insulin signalling by zinc, this micronutrient contributes to calcium signal transduction [44,112].

Extracellular zinc triggers the release of calcium from intracellular stores in HT29 cells. This is mediated by the activation of phospholipase-C (PLC) in response to extracellular zinc binding to G-coupled zinc-sensing receptors on the cell surface. This activation leads to the breakdown of PIP2, which generates two important second messengers, inositol 1,4,5-trisphosphate (IP3) and diacylglycerol (DAG), to promote calcium release from intracellular stores [112]. Like extracellular zinc, intracellular zinc also stimulates calcium flux. Intracellular zinc binding to ryanodine receptor 2 (RYR2) on the sarcoplasmic reticulum triggers calcium release to the cytoplasm in cardiac muscle [44]. Then, calcium flux induced by zinc mediates its downstream events in the cells.

Although studies performed on the mechanism of action of capsaicin and zinc indicate that the pathways activated by these two bioactive food components are distinct, the effectiveness of both capsaicin and zinc in the stimulation of calcium signal transduction suggests this is a common pathway between these molecules.

## 5. Calcium Signalling: The Potential Common Pathway between Capsaicin and Zinc

Calcium is a crucial micronutrient and second messenger responsible for a variety of cellular processes, including cell proliferation, differentiation, migration, gene transcription, and cell death [113]. The balance between calcium distribution through calcium-permeable membrane channels (including TRPV channels) and transmembrane channels is needed for proper calcium signalling [45], and dysregulated calcium signal transduction is associated with many medical conditions including IR and T2DM [114]. Calcium signalling regulates glucose metabolism by stimulation of GLUT4 movement and, eventually, glucose uptake through insulin-dependent and independent signalling pathways [115,116]. Insulin mediates calcium flux from both extracellular space and ER in the cells. Insulin provokes calcium release from ER through the activation of inositol 1,4,5-trisphosphate receptors (IP3Rs), which mediate calcium release from ER. Furthermore, it has been demonstrated that the calcium channel inhibitor 2-APB inhibits the basal rate of calcium influx and reduces insulin-dependent GLUT4 translocation and glucose uptake in the cells. Interestingly, decreased insulin-mediated glucose uptake in skeletal muscle by 2-APB occurs regardless of the activation status of AKT, which suggests that calcium signal transduction mainly affects the later events of the insulin signalling pathway [115].

Calcium exerts its beneficial effect on glucose homeostasis by activation of calcium signalling protein kinases including CAMKKs and AMPK. Calcium signal transduction also regulates the activity of multiple transcription factors [45,48,49]. CAMKK2 is an upstream regulator of CREB, activating the transcription factor by phosphorylation of its serine 133 [117]. Then, CREB recruits its coactivators, TORCs, which are required for full CREB activation. Among TORCs, TORC1 is an important protein in CREB activation and the transcription of the subsequent gene [118]. Moreover, CAMKK2, through dephosphorylation and activation of TORCs, causes full activation of CREB and the expression of downstream CREB target genes [117]. CAMKK2/AMPK and AKT signalling, activated by capsaicin and zinc, respectively, are distinct; however, these two pathways meet each other at CREB downstream of calcium signal transduction, which, perhaps, is the common pathway between capsaicin and zinc in glucose metabolism in skeletal muscle (Figure 4) [33,47,119].

Recent work has shown that the separate treatment of capsaicin and zinc induces an elevation in intracellular calcium levels in mouse skeletal muscle cells. Moreover, this research revealed the critical role of increased intracellular calcium levels in the activation of signalling proteins, including CAMKK2, CREB, and TORC1, which eventually leads to glucose uptake in these cells [33].

## 6. Discussion and Conclusions

Studying cell signalling involved in glucose metabolism could help to identify new therapeutic targets for diseases associated with abnormal glucose metabolism. Bioactive food components are proposed to have beneficial impacts on human health through the activation of cell signalling events. Many attempts have been made to investigate the effectiveness of capsaicin and zinc as valuable bioactive food components in the improvement of glucose homeostasis [23,105,106,120,121,122,123]. Subsequently, several studies have been conducted to find novel clinical therapies for the control of diabetes using zinc and capsaicin [23,86,123,124,125]. For example, zinc supplementation is beneficial in the regulation of blood glucose levels and control of insulin resistance in patients with prediabetes [124] and for glycaemic control in diabetic patients [105]. Like zinc, capsaicin-containing chilli supplementation has been shown to improve postprandial hyperglycaemia and hyperinsulinemia in women with gestational diabetes mellitus [123]. Moreover, regular consumption of chilli-containing foods decreases postprandial hyperinsulinemia and hyperglycaemia in healthy individuals [24]. While clinical studies support the beneficial effects of zinc and capsaicin on glucose metabolism, there is no clear indication of the mechanism of action of these bioactive compounds. Understanding their mechanism of action is essential in the development of new therapies for disorders associated with abnormal glucose metabolism. Therefore, there is a need to study molecular pathways activated by zinc and capsaicin that could lead to the development of novel therapeutic targets for IR and T2DM.

Research suggests that zinc and capsaicin stimulate the activation of the signalling events associated with maintaining glucose homeostasis in target tissues including skeletal muscle [28,29]. Zinc plays an insulin-mimetic role in glucose metabolism by stimulation of key molecules implicated in insulin signalling, including AKT, ERK 1/2, and SHP2, in skeletal muscle cells, independent of insulin action [29,33,94]. This suggests the crucial role of normal zinc homeostasis in maintaining glucose metabolism when there is a deficient level or action of insulin [92,105]. Like zinc, studies have demonstrated the efficacy of capsaicin in the stimulation of glucose uptake through the activation of AMPK, independent of insulin action; however, unlike zinc, capsaicin does not have any impact on the stimulation of insulin signalling molecules such as AKT [28,83]. Furthermore, it has been demonstrated that both capsaicin and zinc stimulate glucose uptake by regulating the activity of calcium/calmodulin-dependent pathways in skeletal muscle [33].

Considering the evaluated impacts of zinc and capsaicin on the control of glucose metabolism, these bioactive components are valuable molecules in insulin-independent glucose uptake. Utilising alternative pathways activated by zinc and capsaicin, which stimulate glucose uptake independently of insulin, could potentially improve glycaemic control in T2DM individuals. However, further work on the mechanism of action of zinc and capsaicin in glycaemic control is required. For example, the expression of TRPV1 on metabolically active tissues such as skeletal muscle makes capsaicin-induced activation of this cation channel an interesting target in drug design for glucose metabolism abnormalities associated with IR and T2DM. ZIP7 is also implicated in the regulation of several genes and proteins involved in glucose homeostasis, which suggests that this zinc transporter could be an effective target for the improvement of glucose metabolism in disorders associated with abnormal glucose homeostasis. Therefore, further work to investigate the impact of TRPV1 and ZIP7 on cellular processes involved in glucose metabolism would help in the development of therapeutic agents targeting these proteins. For instance, enhancing or inhibiting TRPV1 and ZIP7 expression by their transfection could elucidate the function of these genes in the expression of key signalling molecules such as AMPK, CAMKK, AKT, and TORCs, as well as the glucose uptake rate in the cells.

In conclusion, there has been an increasing interest in applying bioactive food components for the treatment of disorders associated with abnormal glucose metabolism, including IR and T2DM. Research on signalling events activated by capsaicin and zinc in glucose metabolism has demonstrated that they improve glucose metabolism by activating signalling pathways, including AMPK, AKT, and calcium signal transduction, independent of insulin in skeletal muscle. Although many attempts have been made to understand signalling pathways activated by capsaicin and zinc in glucose metabolism, several signalling events remain uncovered. Therefore, further study of the mechanism of action of capsaicin and zinc on glucose metabolism will require enhanced knowledge of the molecular mechanisms underlying glucose homeostasis in the normal, IR, and T2DM tissue, as well as the development of novel therapies for these disorders.

## Figures and Tables

**Figure 1 molecules-28-02861-f001:**
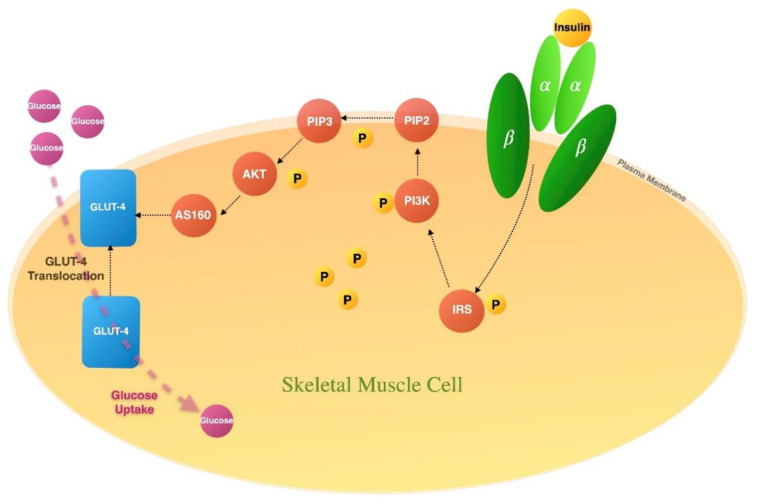
Insulin signal transduction. Insulin binding to its receptor’s extracellular α subunit causes a conformational change in the β subunit of the receptor, which eventually triggers phosphorylation (designated as P) of intracellular proteins, IRSs, which further activates insulin-mediated downstream events including phosphorylation of signalling proteins PI3K, PIP3, AKT, and AS160, leading to GLUT4 movement and glucose disposal.

**Figure 2 molecules-28-02861-f002:**
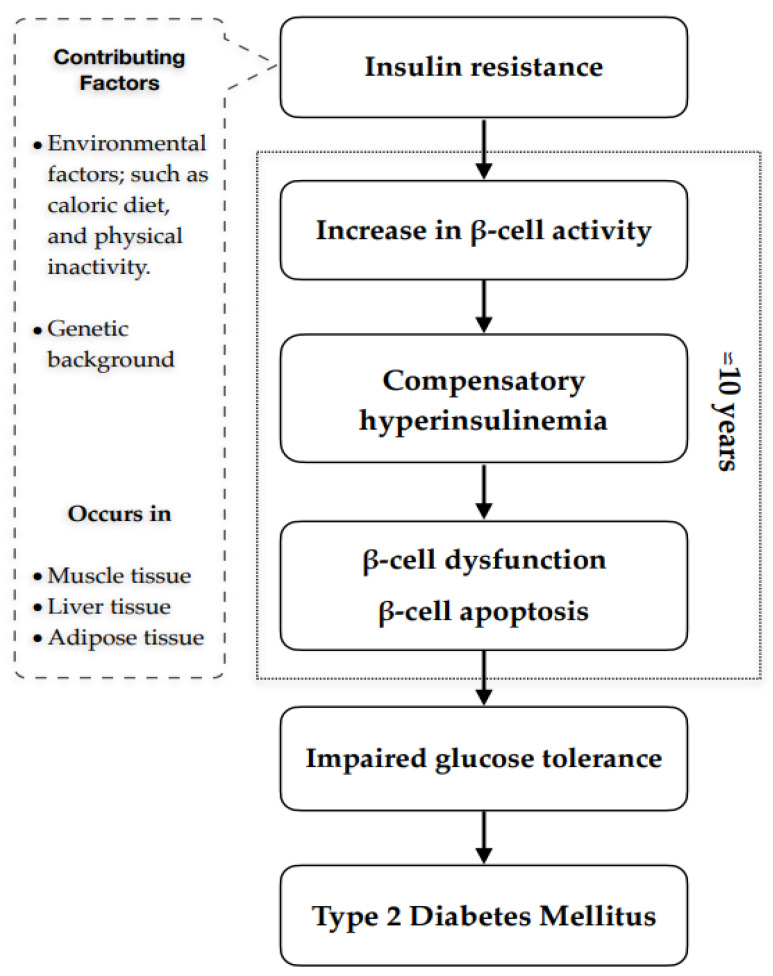
Pathophysiology of T2DM. Several factors can lead to the development of Type 2 Diabetes Mellitus. These include environmental factors such as lifestyle (physical inactivity, diet, etc), and genetics. These factors can contribute to insulin resistance in muscle, adipose, and liver. Insulin resistance is the precursor to beta cell dysfunction, impaired glucose tolerance, and the development of Type 2 Diabetes Mellitus.

**Figure 3 molecules-28-02861-f003:**
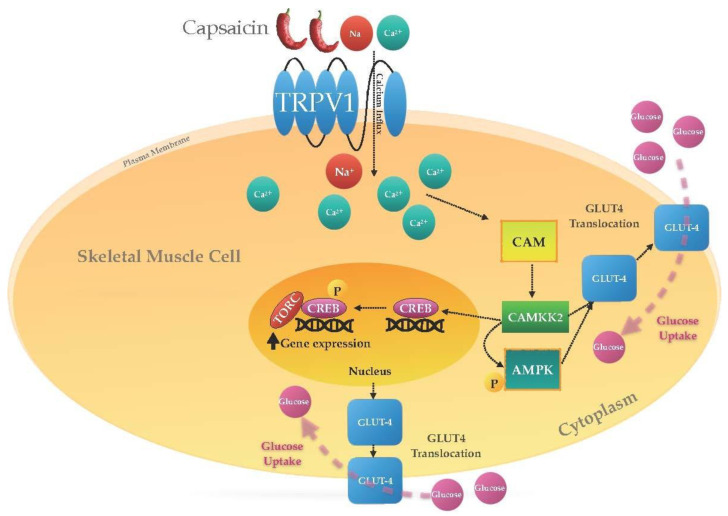
CAMK pathway and the downstream signalling events following TRPV1 activation by capsaicin. Activation of the TRPV1 channel by capsaicin triggers an elevation in calcium flux to the cytosol, which initiates calcium signal transduction in the skeletal muscle cell. Calcium binding to CAM induces a structural modification in CAM and forms a calcium/calmodulin complex. This complex controls the activity of CAMKKs and AMPK, which accordingly stimulate signalling events leading to glucose uptake, including CREB and TORCs activation, and the translocation of Glut4 receptors.

**Figure 4 molecules-28-02861-f004:**
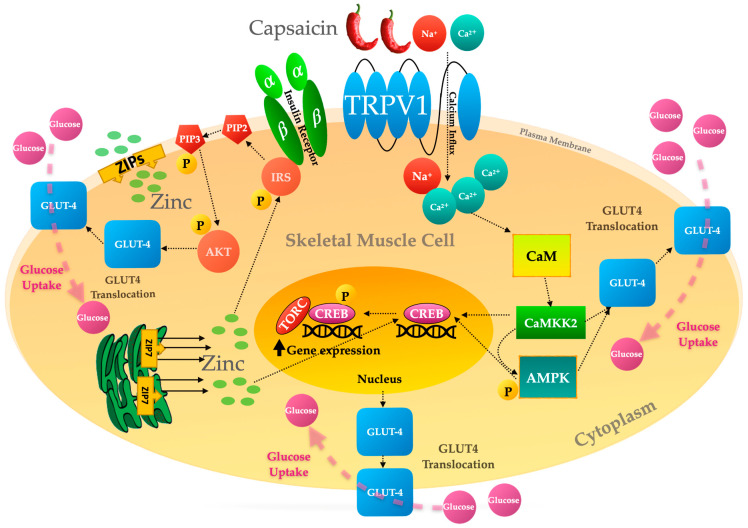
Capsaicin and zinc-induced activation signalling events lead to GLUT4 translocation and glucose uptake in skeletal muscle.

**Table 1 molecules-28-02861-t001:** The tissue-specific distribution of zinc transporters involved in glucose metabolism and the disorders associated with their dysfunction.

Zinc Transporter	Tissue Expression	Disease Associated with Zinc Transporter Dysfunction	Reference
SLC30A3/ZnT3	Brain, testis, and pancreas	Alzheimer’s disease and diabetes	[31]
SLC30A5/ZnT5	Ubiquitously expressed	Osteopenia and diabetes	[94]
SLC30A7/ZnT7	Retina, small intestine, liver, blood, epithelial cells, spleen, and pancreas	Prostate cancer and diabetes	[94]
SLC30A8/ZnT8	Pancreas	Diabetes	[31]
SLC39A6/ZIP6	Ubiquitously expressed	Breast cancer and diabetes	[96,97,98]
SLC39A7/ZIP7	Ubiquitously expressed	Breast cancer and T2DM	[97,99]
SLC39A8/ZIP8	Ubiquitously expressed	Osteoarthritis (OA) and diabetes	[96,97,98]
SLC39A14/Zip14	Small intestine, liver, pancreas, and heart	Colorectal cancer, hepatocellular cancer, prostate cancer, asthma, and diabetes	[91,97]

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
