# Peer review of "Capsaicin and Zinc Signalling Pathways as Promising Targets for Managing Insulin Resistance and Type 2 Diabetes"

_molecules, 2023, doi:10.3390/molecules28062861_

Round 1

Reviewer 1 Report

Comments and Suggestions for Authors

The review is well written even if part of this is a little bit redundant in scholastic  topic. I suggest to remove figure 1 and to avoid repetitions in the clinical complications introduction.

I would like that authors discuss the role of GLUTs and not only the 4 one in IR status and diabetes.

Title or co-title GLUCOSE METABOLISM is not appropriate. It is molecular signalling and not metabolism that is described by authors. 

It is not clear if the actions shown in figures 3 and 4 are only of skeletal muscle cells.

Author Response

Reviewer 2 comments:

General comments:

Reviewer 2

Comments and Suggestions for Authors

1) The review is well written even if part of this is a little bit redundant in scholastic  topic. I suggest to remove figure 1 and to avoid repetitions in the clinical complications introduction.

Thank you very much for your comments. The authors incorporate the comments from the reviewer to enhance the quality of the manuscript. All the changes have been highlighted for the reviewer’s kind consideration.

Figure 1 graphically demonstrates insulin signal transduction. Throughout the manuscript, insulin-independent signalling events and signal transduction stimulated by capsaicin and zinc are also shown graphically. Therefore, the authors believe that keeping Figure 1 will help maintain consistency throughout the manuscript.

2) I would like that authors discuss the role of GLUTs and not only the 4 one in IR status and diabetes.

We have incorporated information regarding the role of GLUTs, as suggested by the reviewer. Please see lines 129-132.

3) Title or co-title GLUCOSE METABOLISM is not appropriate. It is molecular signalling and not metabolism that is described by authors.

The title has been revised. Please see lines 2 and 3.

4) It is not clear if the actions shown in figures 3 and 4 are only of skeletal muscle cells.

More details have been added. Please see lines 247, 266, and 363.

Reviewer 2 Report

Comments and Suggestions for Authors

Overall, this is an interesting manuscript that reviews the importance of capsaicin and zing in glucose metabolism as well as the signaling pathways involved in this mechanism. However, I found many inaccuracies throughout the text that must be corrected before the final version is accepted.

My suggestions are as follows:

Figure 1 is incomplete and lacks clarity. For instance, it is unclear what the single “P” of phosphorylation means. It is recommended that this figure be revised.

The quality of Figure 2 requires enhancement. Additionally, a figure legend should be incorporated to facilitate better understanding.

There is no insulin-stimulated glucose uptake in the liver, as indicated in lines 41-42. Please revise this.

It is necessary to review the following sentence: “AKT stimulates glucose uptake are still unknown; however, the AKT substrate of 160 kDa (AS160) has been established as an important stimulator of GLUT4 translocation” (Line 147-148) because AS160 is required for the retention of GLUT4 within cells and not for its translocation.

For me, the following sentence is unclear and confusing (lane 217): “permitting calcium release from the extracellular space into the cytoplasm.” Please correct this.

In the following sentence: “Moreover, capsaicin induces insulin secretion in wild-type mice, which was shown to be inhibited in Trpv1 knockout mice [25],” it is necessary that authors cite the original article rather than a review.

The author has incorrectly identified SHP2 (lane 100) as an Src homology region 2 domain (SH2), whereas it is, in fact, a protein tyrosine phosphatase 2 (SHP2). This error requires correction. Furthermore, although the authors mention SHP2 several times in the text, the authors never indicate the significance of this phosphatase in relation to zinc-mediated actions.

It is recommended that the term "Calcium/calmodulin-dependent protein kinases" be abbreviated as CAMKs rather than CAMKKs (line 243), as the latter refers to the kinase of CAMKs. Please revise this.

Figure 3 does not depict a role for CAMKs in glucose uptake, contrary to what the figure legend title suggests. Please revise this.

In the following sentence (lines 324-327): “This activation leads to the breakdown of PIP2, which generates two important second messengers, inositol 1,4,5-trisphosphate (IP3) and diacylglycerol (DAG), which work together to promote calcium release from intracellular stores [112],” it is necessary to note that the calcium release, specifically via activated IP3 receptors, is dependent on IP3 and not on DAG as the sentence suggests. A review and correction of this sentence is required.

In the following sentence (358-360): “Moreover, CAMKK2, through dephosphorylation and activation of TORCs, causes full activation of CREB and the expression of downstream CREB target genes [117],” reference 117, does not correspond to the information mentioned in the paragraph.

Figures 3 and 4 present different information regarding CREB activation by TRPV1. While Figure 3 indicates that CAMKK2 is involved in CREB activation, Figure 4 does not indicate that CAMKK2 is involved.

Author Response

Response to reviewer

The authors would like to thank the reviewer for their fair comments. We have addressed each comment below.

Overall, this is an interesting manuscript that reviews the importance of capsaicin and zing in glucose metabolism as well as the signaling pathways involved in this mechanism. However, I found many inaccuracies throughout the text that must be corrected before the final version is accepted.

My suggestions are as follows:

Figure 1 is incomplete and lacks clarity. For instance, it is unclear what the single “P” of phosphorylation means. It is recommended that this figure be revised.

Thank you. We have amended this in the figure legend. 

The quality of Figure 2 requires enhancement. Additionally, a figure legend should be incorporated to facilitate better understanding.

We have provided a legend for Figure 2. “Several factors can lead to the development of Type 2 Diabetes Mellitus. These include environ-mental factors such as lifestyle (physical inactivity, diet, etc), and genetics. These factors can con-tribute to insulin resistance in muscle, adipose, and liver. Insulin resistance is the precursor to beta cell dysfunction, impaired glucose tolerance, and the development of Type 2 Diabetes Mellitus”.

There is no insulin-stimulated glucose uptake in the liver, as indicated in lines 41-42. Please revise this.

This has now been amended to “IR is identified as an impaired response of the target tissues, primarily, adipose tissue and skeletal muscle, to insulin-stimulated glucose uptake.

It is necessary to review the following sentence: “AKT stimulates glucose uptake are still unknown; however, the AKT substrate of 160 kDa (AS160) has been established as an important stimulator of GLUT4 translocation” (Line 147-148) because AS160 is required for the retention of GLUT4 within cells and not for its translocation.

This has been amended to now read “AKT stimulates glucose uptake are still unknown; however, the AKT substrate of 160 kDa (AS160) has been established as an important stimulator of GLUT4 retention”.

For me, the following sentence is unclear and confusing (lane 217): “permitting calcium release from the extracellular space into the cytoplasm.” Please correct this.

This has been amended to “This causes a conformational change in the channel and the opening of a protein pore, permitting calcium release from the extracellular space to enter into the cytoplasm”.

The author has incorrectly identified SHP2 (lane 100) as an Src homology region 2 domain (SH2), whereas it is, in fact, a protein tyrosine phosphatase 2 (SHP2). This error requires correction. Furthermore, although the authors mention SHP2 several times in the text, the authors never indicate the significance of this phosphatase in relation to zinc-mediated actions.

This has now been amended to read “ Src homology region 2 (SHP2; also known as SH2 domain-containing protein tyrosine phosphatase 2)”.

 It is recommended that the term "Calcium/calmodulin-dependent protein kinases" be abbreviated as CAMKs rather than CAMKKs (line 243), as the latter refers to the kinase of CAMKs. Please revise this.

This has been revised

Figure 3 does not depict a role for CAMKs in glucose uptake, contrary to what the figure legend title suggests. Please revise this.

We have added the following to the figure legend, “This complex controls the activity of CAMKKs and AMPK, which accordingly stimulate signalling events leading to glucose uptake, including CREB and TORCs activation, and the translocation of Glut4 receptors”.

In the following sentence (lines 324-327): “This activation leads to the breakdown of PIP2, which generates two important second messengers, inositol 1,4,5-trisphosphate (IP3) and diacylglycerol (DAG), which work together to promote calcium release from intracellular stores [112],” it is necessary to note that the calcium release, specifically via activated IP3 receptors, is dependent on IP3 and not on DAG as the sentence suggests. A review and correction of this sentence is required.

We can see how this sentence could be confusing. This has now been amended to “This activation leads to the breakdown of PIP2, which generates two important second messengers, inositol 1,4,5-trisphosphate (IP3) and diacylglycerol (DAG), which work together to promote calcium release from intracellular stores”.

In the following sentence (358-360): “Moreover, CAMKK2, through dephosphorylation and activation of TORCs, causes full activation of CREB and the expression of downstream CREB target genes [117],” reference 117, does not correspond to the information mentioned in the paragraph.

This relates to the information contained in the paragraph “Among TORCs, TORC1 is an important protein in CREB activation and the transcription of the subsequent gene”.

The sentence “Moreover, CAMKK2, through dephosphorylation and activation of TORCs, causes full activation of CREB and the expression of downstream CREB target genes”, relates to CAMKK2 activation of TORCs and the subsequent activation of CREB.

Figures 3 and 4 present different information regarding CREB activation by TRPV1. While Figure 3 indicates that CAMKK2 is involved in CREB activation, Figure 4 does not indicate that CAMKK2 is involved.

Thank you. This has now been amended.

Round 2

Reviewer 2 Report

Comments and Suggestions for Authors

After reviewing the new version of the manuscript, I agree with the amendments and modifications made in accordance with my previous observations.